# Astrocytes derived from neural progenitor cells are susceptible to Zika virus infection

Edson Iván Rubio-Hernández[1], Mauricio Comas-García[2,3]*, Miguel Angel Coronado-Ipiña[2], Mayra Colunga-Saucedo[4], Hilda Minerva González Sánchez[5]*, Claudia G. Castillo[1]*

1 Laboratorio de Células Neurales Troncales Humanas, Coordinación para la Innovación y Aplicación de la Ciencia y la Tecnología-Facultad de Medicina, Universidad Autónoma de San Luis Potosí, San Luis Potosí, México, 2 Sección de Microscopia de Alta Resolución, Centro de Investigación en Ciencias de la Salud y Biomedicina, Universidad Autónoma de San Luis Potosí, San Luis Potosí, México, 3 Facultad de Ciencias, Universidad Autónoma de San Luis Potosí, San Luis Potosí, México, 4 Sección de Genómica Médica, Centro de Investigación en Ciencias de la Salud y Biomedicina, Universidad Autónoma de San Luis Potosí, San Luis Potosí, México, 5 Cátedra CONACYT- Centro de Investigación sobre Enfermedades Infecciosas, Instituto Nacional de Salud Pública, Cuernavaca, México

* mauricio.comas@uaslp.mx (MCG); claudia.castillo@uaslp.mx (CGC); hilda.gonzalez@insp.mx (HMGS)

## Abstract

Zika virus (ZIKV) was first isolated in 1947. From its isolation until 2007, symptoms of ZIKV-caused disease were limited (e.g., fever, hives, and headache); however, during the epidemic in Brazil in 2014, ZIKV infection caused Guillain-Barré syndrome in adults and microcephaly in fetuses and infants of women infected during pregnancy. The neurovirulence of ZIKV has been studied using neural progenitor cells (NPCs), brain organoids, neurons, and astrocytes. NPCs and astrocytes appear to be the most susceptible cells of the Central Nervous System to ZIKV infection. In this work, we aimed to develop a culture of astrocytes derived from a human NPC cell line. We analyze how ZIKV affects human astrocytes and demonstrate that 1) ZIKV infection reduces cell viability, increases the production of Reactive Oxygen Species (ROS), and results in high viral titers; 2) there are changes in the expression of genes that facilitate the entry of the virus into the cells; 3) there are changes in the expression of genes involved in the homeostasis of the glutamatergic system; and 4) there are ultrastructural changes in mitochondria and lipid droplets associated with production of virions. Our findings reveal new evidence of how ZIKV compromises astrocytic functionality, which may help understand the pathophysiology of ZIKV-associated congenital disease.

## Introduction

Zika virus (ZIKV) is a positive-sense single-stranded RNA [(+)ssRNA] virus that belongs to the *Flaviviridae* family and *Flavivirus* genus [1, 2]. ZIKV was first isolated, in the Zika forest in Uganda in 1947 from a *Rhesus* macaque that developed a febrile illness [3]. Since its isolation and until 2007, only a dozen cases were reported world-wide. However, in 2007, 2013, and 2014 large outbreaks occurred on Yap Island (Micronesia), French Polynesia, and Brazil, respectively. Disease symptoms were similar to those caused by other arboviruses such as dengue virus (DENV), including fever, rash, and headache [4–6]. However, during the epidemic

**Data Availability Statement:** All relevant data are within the paper.

**Funding:** The author(s) received no specific funding for this work.

**Competing interests:** The authors have declared that no competing interests exist.

in Brazil in 2014 ZIKV infection was associated to more severe symptoms, such as Guillain-Barre Syndrome in adults and congenital malformations, fetal death and microcephaly in fetuses and infants of women infected by ZIKV during pregnancy [1].

Currently, two different genotypes of ZIKV have been identified, the African and Asian lineages [5]. The Asian lineage is the one that spread in America from Brazil and is responsible for the congenital disabilities and Guillain-Barré syndrome [5, 6]. There are many other neurodevelopmental disorders associated with ZIKV infection, *e.g.*, ventriculomegaly, intracranial calcifications and ventricular atrophy, but microcephaly is the most common [7, 8].

It is now well recognized that ZIKV infection impairs the development of the Central Nervous System (CNS) [1, 9, 10]. In pregnant women, these neurodevelopmental disorders are more prevalent when infection happens during the first trimester of pregnancy [11, 12]. The effects of ZIKV infection on the CNS have been studied *in vitro* using neural progenitor cells (NPCs) [13–19], brain organoids [14, 19–21], neurons [22–25], and astrocytes [26–30]. In most cases, NPCs and astrocytes are the cells more susceptible to infection by ZIKV [31]. In these cellular models, ZIKV infection compromises cell proliferation [32], induces cell death [15–17], and alters differentiation [13, 14, 33, 34].

Astrocytes are the most abundant cells in the brain, they support and maintain neuron function, contribute to homeostasis, and are a crucial defense mechanism against pathogens [35, 36]. ZIKV infection compromises their viability [28, 30], induces mitochondrial dysfunction [27], and favors an inflammatory state [29, 37]. However, we know significantly less about ZIKV infection of astrocytes than of NPCs and neurons. A significant limitation in the analysis of ZIKV infection in human astrocytes is the lack of non-cancerous cultures. Furthermore, those astrocytes available have not been properly characterized (*e.g.*, their karyotypes, mutational burden, signaling pathways affected, clonal heterogeneity, etc.).

The human neural progenitor cell line (hNS-1) is an attractive alternative to generate astrocyte cultures. It was isolated from a 10.5-week-old human neural fetal tissue that was immortalized with a retroviral vector encoding a v-myc fusion protein [38]. hNS-1 cells proliferate in the presence of basic Fibroblast Growth Factor (bFGF) and Epidermal Growth Factor (EGF) [38–40]. Under these conditions, hNS-1 cells are multipotent and proliferate in a no-differentiated state. After the removal of these growth factors they readily differentiate into neurons, astrocytes, and oligodendrocytes, leaving a remnant of no-differentiated neural progenitors. Moreover, hNS-1 cells have a stable karyotype and do not have major structural modifications.

In this study, we generated human astrocyte cultures from hNS-1 cells (astrocytes-hNS1), characterized by the expression of *bona fide* astrocyte markers: glial fibrillary acidic protein (GFAP), the excitatory amino acid transporters type 1 and type 2 (EAAT1 and EAAT2), and glutamine synthetase (GS). We infected astrocytes-hNS1 cultures finding that ZIKV lowers cell viability, increases the production of Reactive Oxygen Species (ROS), and yields high viral titers. We show that ZIKV modifies the expression of genes essential for CNS homeostasis, such as GFAP, EAAT1, GS, and the N-Methyl-D-Aspartate Receptor (NMDA$_R$), and also of genes that facilitate viral entry. Thin-section Transmission Electron Microscopy (TEM) analysis of infected cells, shows viral factories and the ultra-structural modifications that result in the cytopathic effect of the infected cells. These findings significantly increase our understanding of the pathogenic mechanisms of ZIKV infection in the CNS and also provide the foundation of a model for further characterization.

## Methods

### Propagation of the neural progenitor hNS-1 cells

The human neural progenitor cell line (hNS-1) can be differentiated into neurons, astrocytes, and oligodendrocytes by modifying the proliferation media according to previously published

protocol [38]. Briefly, the hNS-1 cells were cultured in poly-L-lysine-treated culture dished (Sigma-Aldrich, St. Louis, MO) with DMEM-F12 culture medium (Gibco, Invitrogen Life Technologies, Carlsbad, CA) supplemented with 6% glucose, NDiff Neuro-2 Medium Supplement 1X (Merck-Millipore, Darmstadt, Germany), AlbuMAX I 0.5% (Gibco), basic Fibroblast Growth Factor (bFGF, R&D Systems) and Epidermal Growth Factor (EGF, R&D Systems). This cell line was donated by Dr. Alberto Martinez-Serrano at the "Severo Ochoa" Molecular Biology Center, Madrid, Spain.

## Isolation of human astrocytes

The hNS-1 cells were differentiated for 21 days, by removing the growth factors and by adding 0.5% Fetal Bovine Serum (FBS, Sigma-Aldrich), then they were harvested using trypsin with 0.5% EDTA (Sigma) and seeded in culture dishes not treated with poly-L-lysine [41, 42]. The culture was maintained with DMEM Medium (Gibco) supplemented with 10% FBS, 100 U/ml Penicillin, 100 μg/mL Streptomycin, and non-essential amino acids (110 μM L-Alanine, 100 μM L-asparagine, 100 μM L-aspartic acid, 100 μM L-glutamic acid and 100 μM proline). The medium was replaced daily to remove non-adherent cells (*e.g.*, neural progenitors and neurons), and the cells were harvested when they reached a 95% confluency. The generated cultures were designated astrocytes-hNS1; these had at least five passages in which non-adherent cells were no longer observed using bright field microscope with a 10x objective. The cultures were maintained at 37˚C, 95% humidity, and 5% $CO_2$.

## Characterization of astrocytes-hNS1

**GFAP immunofluorescence.** The identity of the astrocytes-hNS1 was confirmed by immunofluorescence against GFAP, a marker routinely used to identify glial cells [42]. Astrocytes-hNS1 were seeded in 24-well plates with coverslips at $6x10^4$ cells per well. After three days, the culture medium was removed, and the samples were recovered for immunofluorescence as previously described [40, 43]. Briefly, cells were washed with phosphate buffer (PBS, 138 mM NaCl, 3 mM KCl, 8.1 mM $Na_2HPO_4$, and 1.5 mM $KH_2PO_4$) and fixed with 4% paraformaldehyde (PFA) for 10 minutes at room temperature. Then the cells were washed three times with PBS, treated with 0.05% sodium borohydride (Sigma-Aldrich) for 10 minutes, washed three times with 0.25% Triton X-100 (Sigma-Aldrich) in PBS, and blocked with 20% Horse Serum (Gibco) in PBS for 2 hours. Subsequently, the cells were incubated with the primary rabbit anti-GFAP antibody (1:1000, Cat. No. Z033401, Dako) overnight at 4˚C, cells were then washed twice with 0.25% Triton X-100 1% and horse serum in PBS and incubated with the secondary antibody (Alexa-546 goat anti-rabbit a 1:300 dilution, Cat. No. A-11010, Invitrogen) for 2 hours at room temperature. The nuclei were stained with Hoechst 33258 (0.2 μg/ mL, Invitrogen) for 10 minutes, washed four times with PBS, and the coverslips were mounted with Vectashield mounting medium (Vector labs). The cells were analyzed in the LEICA DM2500 epifluorescence microscope with the filter cube A (BP:340–380), cube I3 (BP:450–490, 510), and cube N2.1 (BP:515–560, 580). A total of 7–8 fields per coverslip were captured, 3–4 coverslips were analyzed per independent experiment, and four independent experiments were performed.

## RT-PCR of GFAP

Onto 6-well plates, $5x10^5$ cells per well were seeded and allowed to proliferate for three days. The total cellular RNA was obtained on the third day using Trizol^TM (Life Technologies, Rockville, MD), following the supplier's instructions. The expression of the glial marker *GFAP*, was determined by one-step endpoint RT-PCR using the OneStep RT-PCR kit (Qiagen, Cat. No.

210212) according to the supplier's instructions. Using 50 ng of RNA in a reaction volume of 25 μL, PCR was performed under the following conditions: Reverse transcription at 50˚C for 30 minutes; activation of DNA polymerase at 95˚C for 15 minutes; 30 cycles of denaturation at 94˚C for 30 seconds, alignment at 60˚C for 30 seconds, and extension at 72˚C for 1 minute; and a final extension at 72˚C for 10 minutes. The amplification products were visualized on a 1% agarose gel. The sequences of the primers used, and the size of the amplicon are depicted in S1 Table.

## Analysis of the karyotype

The karyotype of the astrocytes-hNS1 was analyzed to determine their genomic integrity in culture. Briefly, $2.5 \times 10^5$ cells per well were seeded in 6-well plates and allowed to proliferate for two days. The cells were washed with PBS and incubated with colchicine (1 μg/mL) for 2 hours at 37˚C, then the colchicine was removed, and the cells were washed once with PBS and harvested with trypsin. The cells were centrifuged at 1500 rpm for 5 minutes, and the pellet was resuspended in 1.5 mL of fresh medium, 10 mL of warmed hypotonic solution (75 mM KCl) were added dropwise and the cells were incubated for 20 minutes at 37˚C. The cells were centrifuged at 2000 rpm for 10 minutes, the supernatant was eliminated, leaving 1 ml behind and flicking the pellet. Next, 10 ml of ice-cold fresh fixative solution (75% methanol and 25% acetic acid) was added, and the cells were centrifuged for 5 minutes at 2000 rpm. This step was repeated twice. The cellular pellet was resuspended in 1 ml of fixative solution and poured onto cold slides. The sample was allowed to dry at room temperature, and fixation was performed using heat. The next day the samples were stained for 10 minutes with 0.4% Giemsa (Sigma-Aldrich), followed by washes with 4% ethanol in water and pure water. The metaphases were observed under a Motic microscope with the 100X objective.

## Viral propagation, production of stocks and infections

ZIKV infectious viral particles were obtained by transfecting HEK-293T cells with an infectious cDNA that contained a full-length clone of ZIKV isolated from the 2015 epidemic in Brazil [44]. The ZIKV-ICD plasmid (GenBank: KX576684.1) was donated by Dr. Alexander Pletnev from NIAI/NIH. This plasmid was transfected into HEK-293T cells using the calcium phosphate method [45]. Viral stocks were obtained by infecting Vero E6 cells with the supernatant of the transfected cells. The viral titers were determined by plaque assay using Vero E6 cells as described previously [29]. For infections of the astrocytes-hNS1, cells were seeded, washed twice with serum-free medium, and the viral inoculum was set at the desired multiplicity of infection (0.1 and 1 MOI). The inoculum was incubated for 2 hours, shaking every 10 minutes prior to being removed. The cells were washed once with serum-free medium, and a fresh medium was added to maintain the culture until the required post-infection times.

## Viral production kinetics and percentage of infected cells

Viral production was analyzed by seeding $5 \times 10^5$ astrocytes-hNS1 per well in 6-well plates, and 24 hours later were infected at a MOI of 1. The culture was maintained for five days post-infection (dpi), one-third of the medium was replaced by a fresh medium every 24 hours, and the removed media was stored at -80˚C until quantification of viral load.

To determine the percentage of infected cells, $6 \times 10^4$ cells per well were seeded in 24-well plates and infected at a MOI of 1 at 24 hours post-seeding. At three dpi, the cells were analyzed by immunofluorescence using a mouse anti-flavivirus protein E antibody (1:1000, MAB10216, Merck-Millipore) and Alexa-488-conjugated goat anti-mouse secondary antibody (1:200, Invitrogen). Cells were analyzed with a Lionheart FX microscope (Biotek, Winooski, VT, USA)

with the DAPI filter (Ex 377/50 Em 447/60, Mirror 409), the GFP filter (Ex 469/35 Em 525/39, Mirror 497), and the TRITC filter (Ex 556/20 Em 600/37, Mirror 573). A total of 8 regions of interest were captured per well using the 20X objective, 3–4 wells were analyzed per independent experiment, and four independent experiments were performed. The number of positive cells was determined using the software Gen5 Image Prime 3.03 (Biotek) by automatic and manual detection. The total number of cells in the field was determined by counting the number of nuclei (Hoechst stained); the number of infected cells corresponds to the cells positive to the anti-E antibody; and the percentage of infected cells was obtained by the ratio of nuclei to E-positive cells.

## ZIKV RNA extraction and quantification

The viral RNA from culture supernatants obtained at the dpi of interest was used to determine the viral titers. The viral RNA was isolated using the RNA/DNA purification kit (Magnetic bead) from Da An Gene Co., Ltd. of Sun Yat-Sen University. The absolute quantification was performed using the Prime Script RT-PCR One-Step TB Green II kit (Takara, Bio Inc.) in the 7500 Fast Real-time PCR System (Applied Biosystems). The reaction conditions were 7 minutes at 42°C, 40 cycles of 10 seconds at 95°C, and 1 minute at 60°C, followed by denaturation analysis. The sequences of the primers and the sizes of the amplicon are depicted in S1 Table. The RNA levels were quantified using a standard curve with known amounts of ZIKV-ICD plasmid. A total of three independent experiments were performed, and samples were read in duplicate.

## Determination of metabolic activity and production of reactive oxygen species

Astrocytes-hNS1 were seeded in 96-well plates at a density of $1.5 \times 10^4$ cells/well; the next day, they were infected with ZIKV at 0.1 and 1 MOI. At five dpi, the metabolic activity was analyzed using resazurin (Sigma Aldrich) [40, 46], and the production of ROS was determined using 2',7'-dichlorofluorescein diacetate (DCFH-DA, Cat. No D6883, Sigma Aldrich) [47]. Briefly, the medium was removed and replaced with DMEM medium without phenol red and supplemented with resazurin (10 μg/ mL final concentration), then the cells were incubated for 2 hours at 37°C. The metabolic activity was determined by monitoring the fluorescence of resorufin (resazurin metabolized compound) in a Synergy H1 microplate reader (Biotek) at 560 nm excitation and 590 nm emission. For the analysis of ROS production, the medium was replaced with DMEM without phenol red and supplemented with 10 μM DCFH-DA; the cells were incubated for 2 hours at 37°C. The fluorescence reading was performed at 480 nm excitation and 530 nm emission. For both assays, 3 to 4 independent experiments were performed with three experimental replicates, and the fluorescence emitted by uninfected cells was normalized as 100% metabolic activity and ROS production, respectively.

## Quantification of astrocyte-specific markers and lipid metabolism by RT-qPCR

Changes in the expression of astrocyte-specific markers and genes involved in hemichannel connection between cells and lipid metabolism were analyzed by RT-qPCR. For this, $6 \times 10^5$ cells per well were seeded in 6-well plates, infected with ZIKV (MOI of 1), and at five dpi, the total cellular RNA was obtained using Trizol$^{TM}$ (Life Technologies) following the supplier's instructions. Subsequently, 500 ng of cellular RNA were retrotranscribed with the SuperScript II reverse transcription kit (Invitrogen). The relative expression of the genes of interest was

determined using a two-step RT-PCR with the iQ SYBR Green Supermix kit (BioRad, Hercules, CA) in the 7500 Fast Real-time PCR System (Applied Biosystems). The setup of the reactions was the following: 95˚C for 3 minutes, 40 cycles of 95˚C for 10 seconds, and 60˚C for 1 minute, followed by thermal denaturation analysis. Primers and amplicons are listed in S1 Table. The RNA levels of the genes of interest, as well as of the viral RNA, were normalized using glyceraldehyde-3-phosphate dehydrogenase (GAPDH) expression. The data were analyzed using the $2^{\wedge}$-$\Delta\Delta$Ct method. A total of 6 independent experiments were performed.

## Transmission electron microscopy analysis

The ultrastructural characterization of infected cells was performed by thin-section TEM. For this, $6\times10^5$ cells per well were seeded in 6-well plates and infected with ZIKV (MOI of 1). The cells were fixed at 3 and 6 dpi with 2% glutaraldehyde, 0.1 M sodium cacodylate, and 0.2% picric acid for 2 hours at room temperature and 72 hours at 4˚C. The samples were washed three times with 0.1 M sodium cacodylate pH 7.2, then fixed with 1% osmium tetroxide in 0.1 M sodium cacodylate for 1 hour at room temperature and in the dark, then they were washed twice with the sodium cacodylate buffer, and twice with a 0.1 M sodium acetate buffer pH 4.5. The cells were stained in block with 0.5% uranyl acetate and 0.1 M sodium acetate for 1 hour at room temperature in the dark, washed three times with 0.1 M sodium acetate, and three times with MilliQ water; then, they were dehydrated with ethanol at increasing concentrations (35%, 50%, 70%, 95%, and 100%) for 10 minutes and three times for each concentration. The cells were washed three times with EMbed 812 resin (Electron Microscopy Sciences). Finally, the resin was allowed to polymerize for 48 hours at 55˚C. The polymerized disks were sectioned (70 nm thick sections) using a diamond blade in the LEICA EM UC7 ultramicrotome; the sections were transferred to copper grids and stained with 0.5% uranyl acetate and 0.5% lead citrate in closed Petri dishes containing dry pellets of NaOH. The samples were visualized in the transmission electron microscope JEM-JEOL-2100 at 200 kV, and the images were obtained with the One View Gatan 4K camera. Micrographs were processed in ImageJ software. The number of mitochondria and Lipid Dropets (LDs) per cell were calculated using micrographs at different magnifications and having as reference other cell organelles.

## Statistical analysis

Data were first analyzed to determine their distribution. Data with normal distribution were expressed as mean +/- standard deviation and analyzed with the t-student or ANOVA for multiple comparisons with Bonferroni's posthoc test. Data with non-parametric distribution were expressed as the median and interquartile range (IQR) and analyzed with the U Mann-Whitney or Kruskal-Wallis tests and using Dunn's post hoc test. A $p<0.05$ was considered significant. All data were analyzed and graphed using GraphPad software prism 5.0.

## Results

### Generation of astrocytes-hNS1

Astrocytes are a crucial component of the regulation of neurotransmission [48]. They promote myelination, provide support and protection to neurons, and help to maintain the central and peripheral nervous system homeostasis. Astrocytes are the most abundant component of the macroglia in the CNS. To investigate the effects of ZIKV infection on astrocytes we generated cultures of these cells by differentiating the multipotent progenitor hNS-1 cell line (Fig 1). Twenty-one days after the differentiation protocol is started, the progenitor cell line differentiates into two distinct populations: a GFAP-positive population of astrocytes representing

29.8 ± 7.6% of the cells in culture, and a neuronal population positive for microtubule-associated protein-2 (MAP-2) representing 27.2 ± 7.6% [39, 40]. The other cells in culture remain as undifferentiated hNS-1 cells. Then, the astrocytes are enriched by mechanical and physical exclusion; unlike hNS-1 and neurons, adhere to culture dishes without poly-L-lysine treatment (Fig 1A). The adherent cells were passed three more times to increase the purity of the culture (Fig 1A). After that, cells were counted by immunofluorescence staining observing that 93.0 ± 0.3% were GFAP-positive cells (n = 4) (Fig 1B). In addition, the astrocyte cultures were analyzed for expression of the glial cell marker GFAP by end-point PCR, confirming the positivity of this marker (S1 Fig). Finally, we assessed whether the differentiated astrocytes were genomically stable by analyzing the karyotype of different passages (P10, P15, and P22), observing that the chromosomes remained intact (Fig 1C).

## The astrocytes-hNS1 are susceptible to Zika virus infection

Once the culture of astrocytes-hNS1 was obtained and characterized, we determined whether the cells were susceptible to ZIKV infection (Fig 2). Astrocytes-hNS1 were infected with a MOI of 1, and infection was assessed at three dpi by immunofluorescence staining with an antibody directed against the flavivirus protein E. The average percentage of infected cells was 26% (± 4.5%, n = 6) (Fig 2A and 2B). RNA was extracted from the culture supernatants of infected cells and the viral load was quantified by RT-qPCR. At 1 dpi the viral production was $10^{5.9}$ copies/μL, at 2 dpi the production increased to $10^{6.5}$ copies/μL, and for 3, 4, and 5 dpi the viral production was maintained between $10^{7.2}$ and $10^{7.7}$ copies/μL (Fig 2C). In order to determine if there was still viral genome replication at 5 dpi, the cellular RNA was obtained from infected astrocytes and quantified by RT-qPCR. The amount of viral genome in the cell was $10^{5.8}$ copies/ng of RNA total. Using bright field microscopy, we observed changes in the cell morphology characterized by rounded and refringent cells, accompanied by cells with elongated cell processes, all consistent with a generalized cytopathic effect (see white arrows, Fig 2D). Infection was also confirmed by thin-section TEM of infected cells at 6 dpi, in which we observed the presence of viral particles inside cytoplasmic vesicles (see white arrows, Fig 2E). Altogether these data support that astrocytes-hNS1 are susceptible to ZIKV infection and that at 5 dpi there is still viral genome replication; therefore, they can be a useful model to study the damage that ZIKV causes to cells of the nervous system.

## Infection affects metabolic activity and production of reactive oxygen species

We sought to indirectly analyzed cell death by measuring changes in the metabolic activity of infected cells through resazurin reduction. The metabolic activity of ZIKV-infected cells at five dpi at MOI of 0.1 and 1 was reduced significantly compared with the uninfected control (set at 100%): 84.8% ± 6.6%, p<0.05, and 70.3% ± 4.8%, p< 0.01, respectively (Fig 3A). We evaluated whether the decrease was due to the generation of intracellular ROS using the compound DCFH-DA. Indeed, ROS production was higher in ZIKV-infected astrocytes at five dpi for both, MOI of 0.1 (116% ± 6.5%, p<0.05) MOI of 1 (122.5% ± 5.8%, p<0.01) (Fig 3B), than in the mock infected cells. These data further confirm that astrocytes-hNS1 can be infected by ZIKV, also arguing that infection decreases cell viability due to a decrease in metabolic activity and an increase in oxidative stress.

## Infection alters the expression of glial proteins

To analyze whether ZIKV infection compromises the identity and/or function of differentiating/differentiated glial cells, we infected astrocytes-hNS1 at a MOI of 1, and compared the

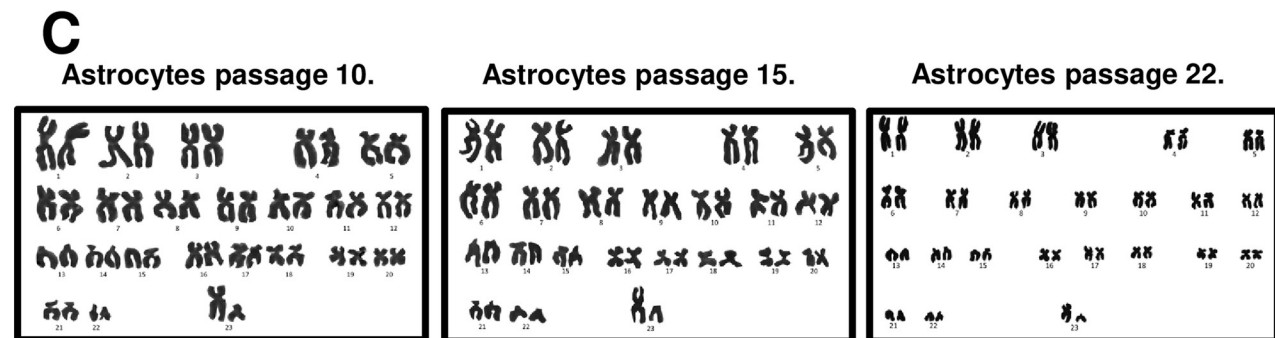

**Fig 1. Astrocytes-hNS1 are a pure culture of human astrocytes.** hNS-1 cells were differentiated for 21 days. The differentiated cells were harvested and seeded in culture labware without poly-L-lysine treatment and with DMEM. Non-adherent cells (neurons and neural progenitor cells that did not differentiate) were removed by exchanging the media. Astrocytes-hNS1 cells were seeded, and cells were fixed and processed for immunofluorescence using antibodies to GFAP (astrocyte marker). Cells from different passages were used to verify that they maintained a stable karyotype using the karyotyping technique. (A) Diagram of the procedure to obtain a pure culture of astrocytes-hNS1. (B) Representative micrographs of GFAP immunofluorescence in cultured astrocytes. Scale bar 50 µm. (C) Reconstruction of the karyotype of three passages of astrocytes-hNS1.

expression of the glial transcripts *GFAP*, *SLC1A3* (encodes EEAT1*)*, *SLC1A2* (encodes EEAT2), *GLUL* (encodes GS*)*, *GRIN1* (encodes N-methyl-D-aspartate Subunit 1 Receptor, $NMDA_R$ *Subunit 1*) and *GJA1* (encodes connexin-43 (CX43)) at 5 dpi. GFAP is a structural

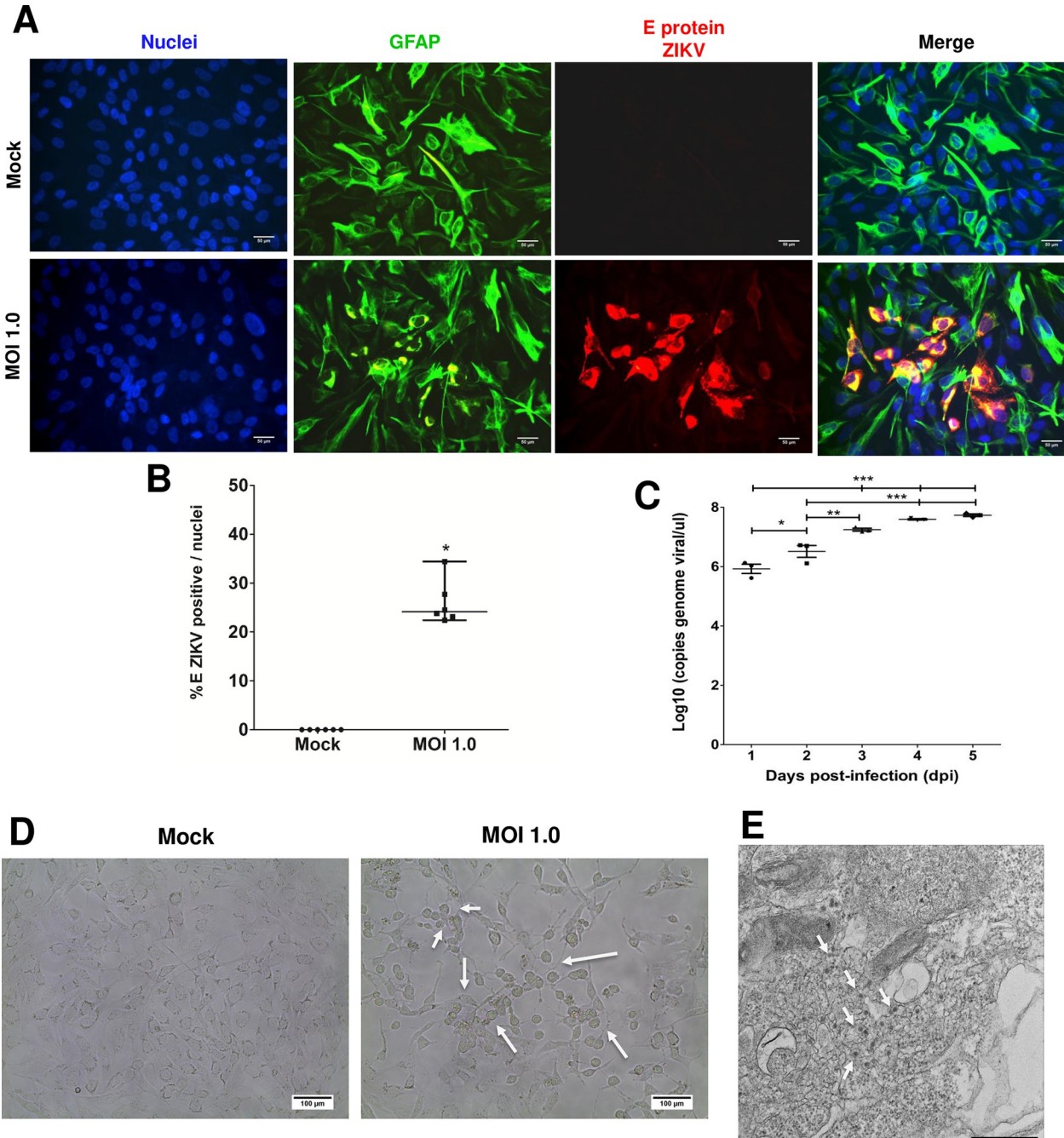

**Fig 2. Astrocytes-hNS1 are permissible and susceptible to ZIKV infection.** (A) The astrocytes-hNS1 were seeded in 24-well plates and processed for immunofluorescence with antibodies against GFAP (green) and ZIKV envelope protein (red), and the nuclei were stained with Hoechst (blue). Scale bar 50 μm. (B) The number of cells positive for ZIKV infection was quantified at three dpi with MOI of 1 (mean percentage and standard deviation, n = 6, p = 0.031, t-test). (C) The Astrocytes were infected with ZIKV (MOI of 1), and supernatants were recovered for 5 dpi to determine viral genome production by qRT-PCR. An increase in the viral genome during the infection progress is shown (*p<0.05, **p<0.01, ***p<0.001, ANOVA with Tukey's multiple comparisons). Each point represents the mean of 3 independent experiments with at least two duplicates, and the error bars correspond to the standard deviation. (D) Brightfield microscopy of ZIKV infected (MOI of 1) or mock-infected astrocytes-hNS1 at 5 dpi. Scale bar 100 μm. White arrows indicate cytopathic effect (E) Thin-section TEM of infected cells (MOI of 1) at 6 dpi shows the presence of viral particles inside vesicles (white arrows).

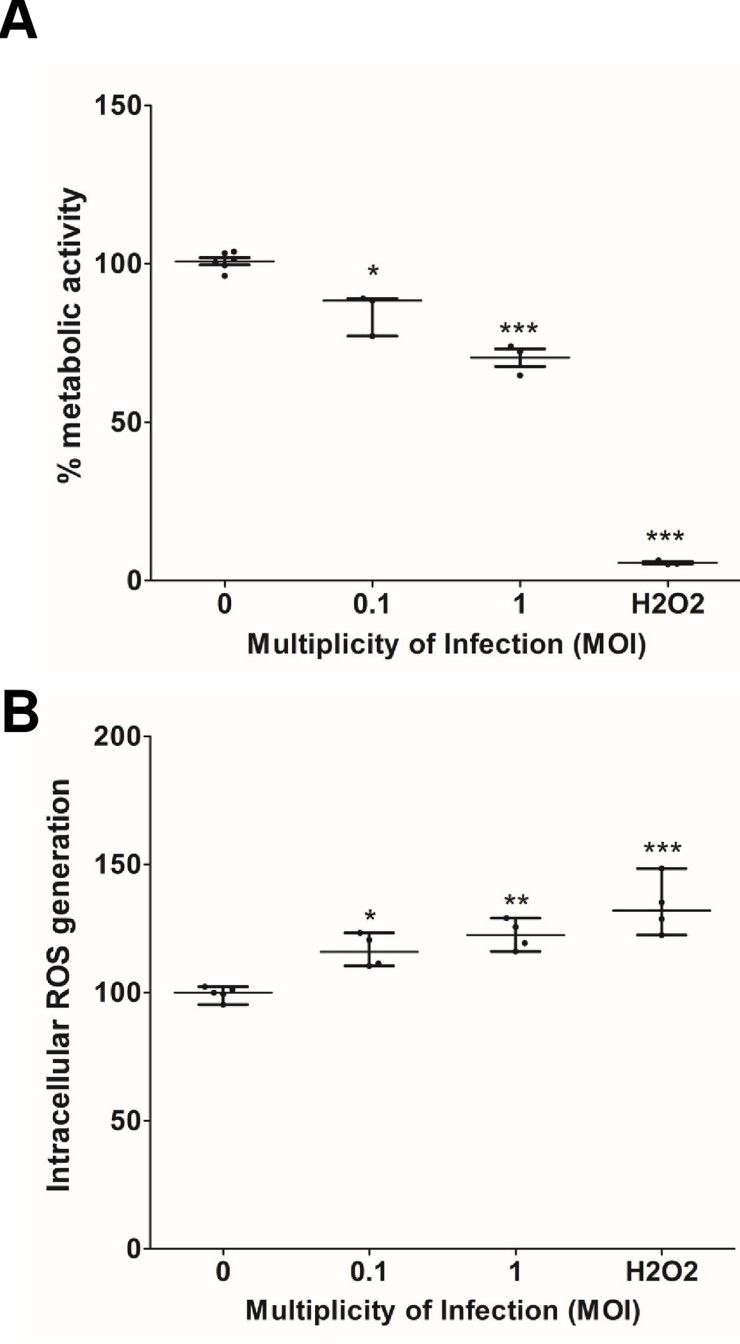

**Fig 3. ZIKV infection of astrocytes modifies the metabolic activity and the production of species reactive oxygen.** (A) Astrocytes-hNS1 were infected with ZIKV (MOI of 0.1 and 1), and at five dpi, viability was determined by indirectly measuring activity metabolism with resazurin. (B) The production of ROS was measured in mock-infected and ZIKV-infected cells at MOI of 0.1 and 1. In A and B, the data are median and interquartile range, n = 3–4, **p<0.01, ***p<0.001, Kruskal Wallis with Dunn's post hoc. For A and B, $H_2O_2$ was used to induce cell death.

protein that helps to maintain the shape of astrocytes, EAAT1 and EAAT2 are glutamate transporters that internalize glutamate in astrocytes and are thus the main regulators of excitatory glutamate neurotransmission. GS synthesizes glutamine using glutamate and ammonia as substrates, and with this protects neurons from the toxicity resulting from the excess of these

compounds during the events of neuron excitation. NMDAR is an ionotropic glutamate receptor and CX43 is a hemichannel protein found in the gap junctions that connect adjacent cells. [49–51].

We observed that the relative expression of GFAP ($0.22 \pm 0.04$, $p < 0.001$), EAAT1 ($0.36 \pm 0.07$, $p < 0.001$), and GS ($0.79 \pm 0.11$, $p = 0.06$) decreased in infected cells compared with control cells (set to 1) (Fig 4A–4C). On the contrary, the relative expression of EAAT2 ($1.35 \pm 0.3$, $p = 0.06$) and Cx43 ($1.62 \pm 0.7$, $p = 0.1$) showed a non-significant trend to increase (Fig 4D and 4E). Finally, the relative expression of the NR1 subunit of NMDA$_R$ was significantly upregulated in the infected cells ($1.78 \pm 0.5$, 1 $p < 0.005$) (Fig 4F).

## Expression of Zika virus entry receptors and genes involved in lipid metabolism

The TAM (Tyro-3, AXL, and Mertk) family of tyrosine kinase receptors are used for viral entry of flavivirus DENV and ZIKV into host cells [52]. We analyzed the expression of *Tyro-3*, *AXL*, and *Mertk* basally and upon ZIKV infection in our hNS1-derived astrocyte cultures. All three genes were expressed in the absence of infection. However, the expression of *Tyro-3* was reduced upon infection compared with the mock-infected cells (set at 1) ($0.63 \pm 0.16$, $p = 0.025$) (Fig 5A). On the contrary *AXL* and *Mertk* expression increased (AXL: $3.25 \pm 0.54$, $p < 0.001$; Mertk: $1.44 \pm 0.21$, $p < 0.05$) (Fig 5B and 5C, respectively).

The lipidic component of the cellular membrane also influences flavivirus infection, particularly the amount of cholesterol [53, 54]. We analyzed the expression of the genes encoding the peroxisome proliferator-activated receptor-gamma (PPAR-$\gamma$) and apolipoprotein E (APOE) since both are involved in lipid homeostasis. The relative expression of *PPAR-$\gamma$* ($p = 0.39$) and *APOE* ($p = 0.24$) in cells infected with ZIKV (MOI of 1) was not modified (Fig 5D and Fig 5E). These data further support the ability of ZIKV to infect astrocytes-hNS1 through TAM family receptors.

## Zika virus replication is localized to membranous organelles

To further understand the infection process of hNS1-derived astrocytes, infected cells were analyzed by thin-section TEM. Fig 6A shows a representative micrograph of an entire astrocyte-hNS1. Cells are characterized by large multilamellar bodies (see blue arrows), which are lipid-rich organelles that contain cytoplasmic material. Fig 6B shows an infected cell at 6 dpi (MOI of 1), in which it can be observed the presence of vacuoles that occupy most of the cytoplasmic space (see red arrows). These vacuoles can be empty or contain smaller vesicles. In Fig 6A we point out to a multilamellar body with blue arrows; however, these structures were very rare in infected cells and were small (see Fig 6B), contrary to uninfected cells. In Fig 6C and 6D we point out to two different cells with increased levels of viral production. The cell in Fig 6C has only one vesicle that contains viruses (see green arrows); at first glance this is the only modification but as it will be discussed later there is an abnormal mitochondrion, and the rough endoplasmic reticulum (RER) is not visible. On the contrary, the cell in Fig 6D shows a completely vacuolized cytoplasm, with most of the small vesicles containing viral particles and a RER completely swollen. These two cells may represent different stages of viral infection.

## Zika virus infection modifies the appearance of mitochondria

The increase in ROS production suggests possible ZIKV-induced mitochondrial damage. Thus, we further characterize this organelle by thin-section TEM. The morphology of the mitochondria in the mock-infected cells is as expected (*e.g.*, well-defined, and with parallel cristae) (Fig 7A and 7B). However, in ZIKV-infected cells the mitochondria exhibit damage,

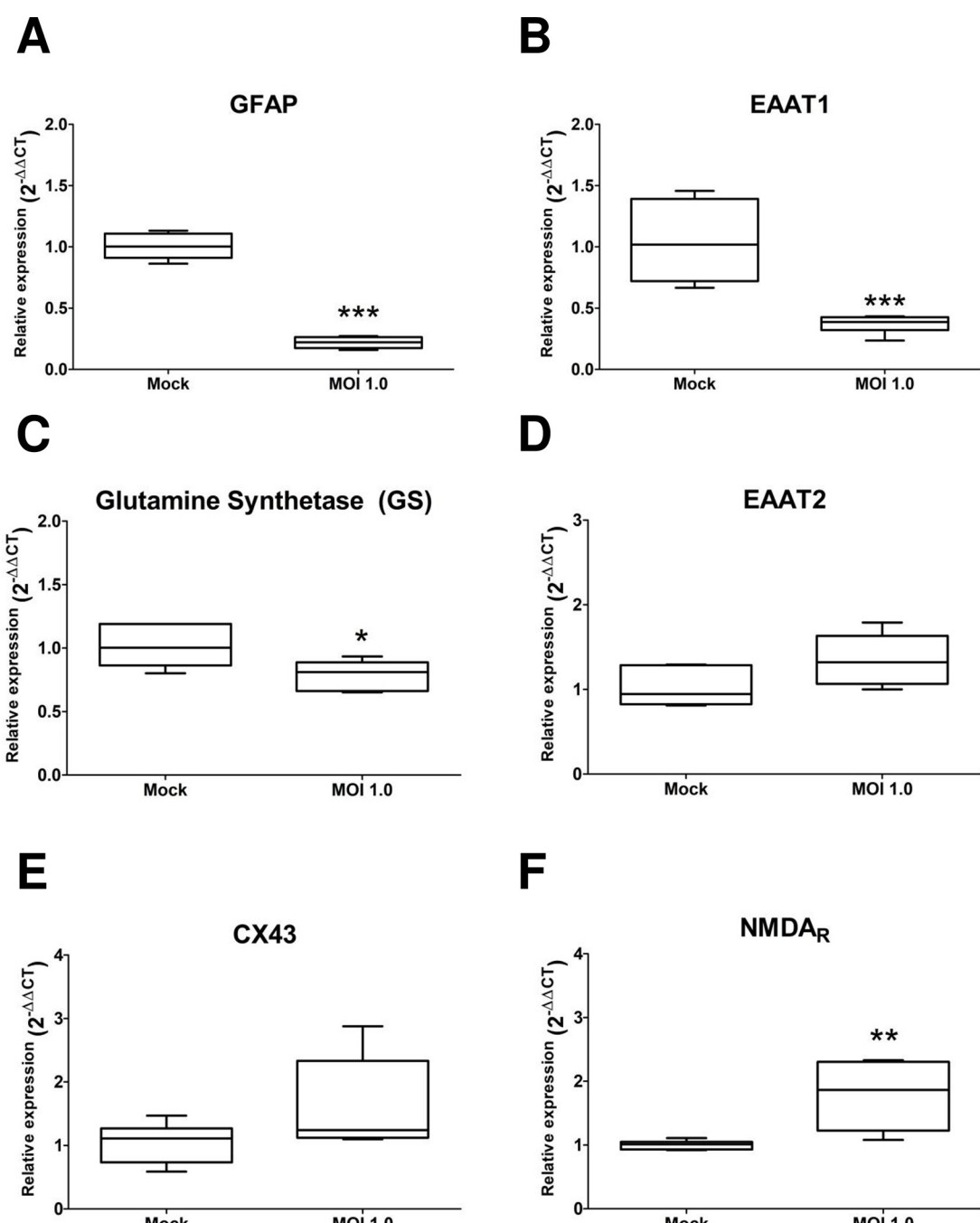

**Fig 4. Glial marker expression is altered by ZIKV infection.** The astrocytes-hNS1 culture was infected with ZIKV (MOI of 1) or mock-infected. At five dpi the cellular RNA was recovered and the expression of glial transcripts was analyzed by qRT-PCR. (A) Relative expression of GFAP in infected and uninfected cells, p = 0.002, n = 6, t-test. (B) Relative expression of EAAT1, p = 0.0006, n = 6, t-test. (C) Relative expression of GS, p = 0.023, n = 6, t-test. (D) Relative expression of EAAT2, p = 0.0006, n = 6, t-test. (E) Relative expression of CX43, p = 0.10, n = 6, U Mann Whitney test. (F) Relative expression of NMDA$_R$, p = 0.0045, n = 6, t-test. The plotted data represent the mean and standard deviation.

and this damage seems to depend on the progression of infection. At 3 dpi, mitochondria exhibit concentric cristae, and the external membrane is swollen (Fig 7C). By day 6, mitochondria remain swollen and there is a decrease in the thickness and the number of cristae

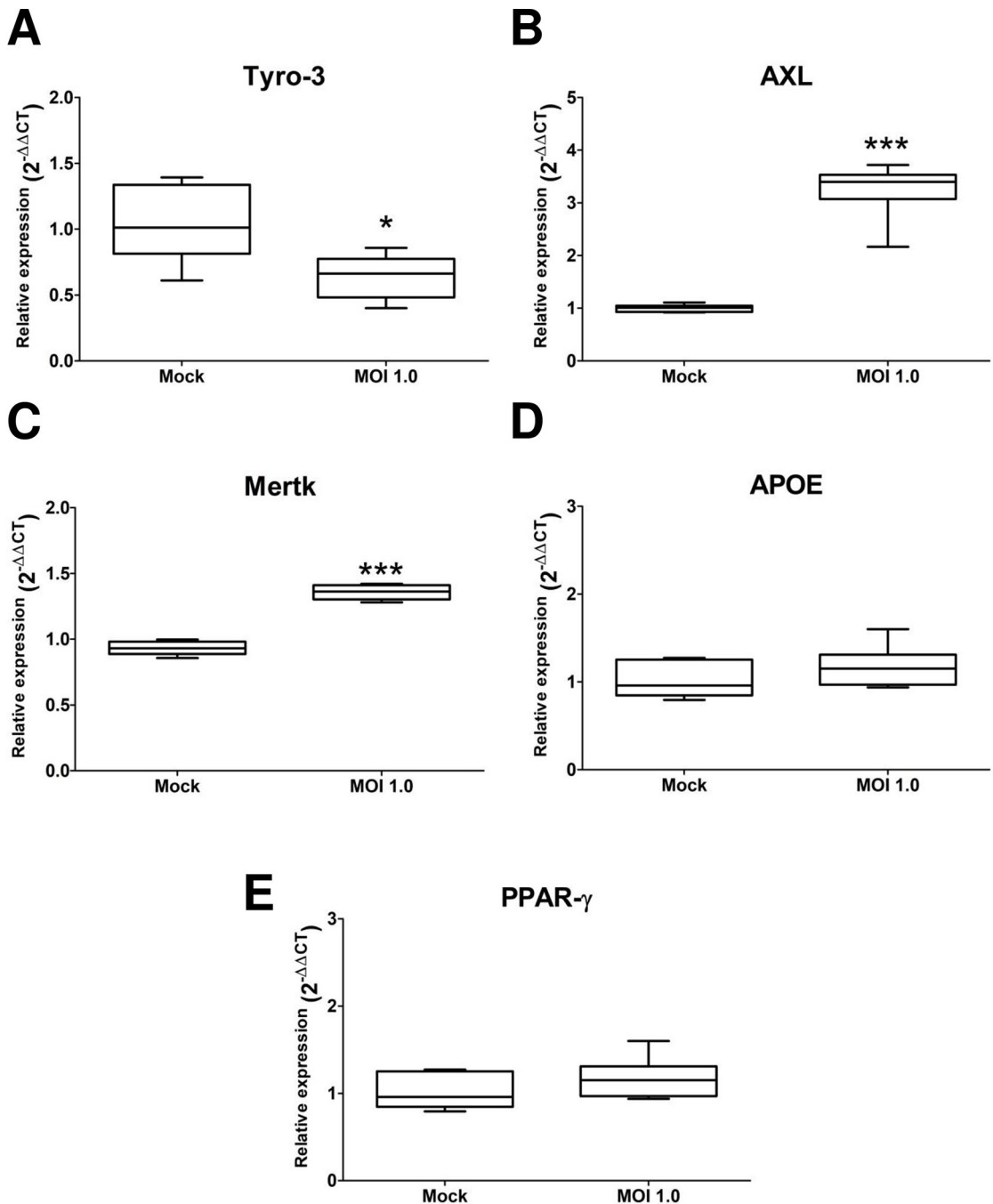

**Fig 5. ZIKV infection can modify genes of the TAM family and genes involved in the lipidic metabolism.** The astrocytes-hNS1 culture was infected with either ZIKV (MOI of1) or mock-infected. At five dpi, the cellular RNA was recovered and the relative expression of the mRNAs for the receptors of the TAM family (Tyro-3, AXL, and Mertk), and APOE and PPARγ mRNAs associated with lipid metabolism were determined by qRT-PCR. (A) Tyro-3 relative expression, p = 0.025, n = 6, t-test. (B) Relative expression of AXL, p = 0.0022, n = 6, U Mann Whitney test. (C) Relative expression of Mertk, p = 0.0006, n = 6, t-test. (D) Relative expression of PPAR-γ, p = 0.24, n = 6, t-test. (E) APOE relative expression p = 0.18, n = 6, t-test. The data correspond to the mean and standard deviation.

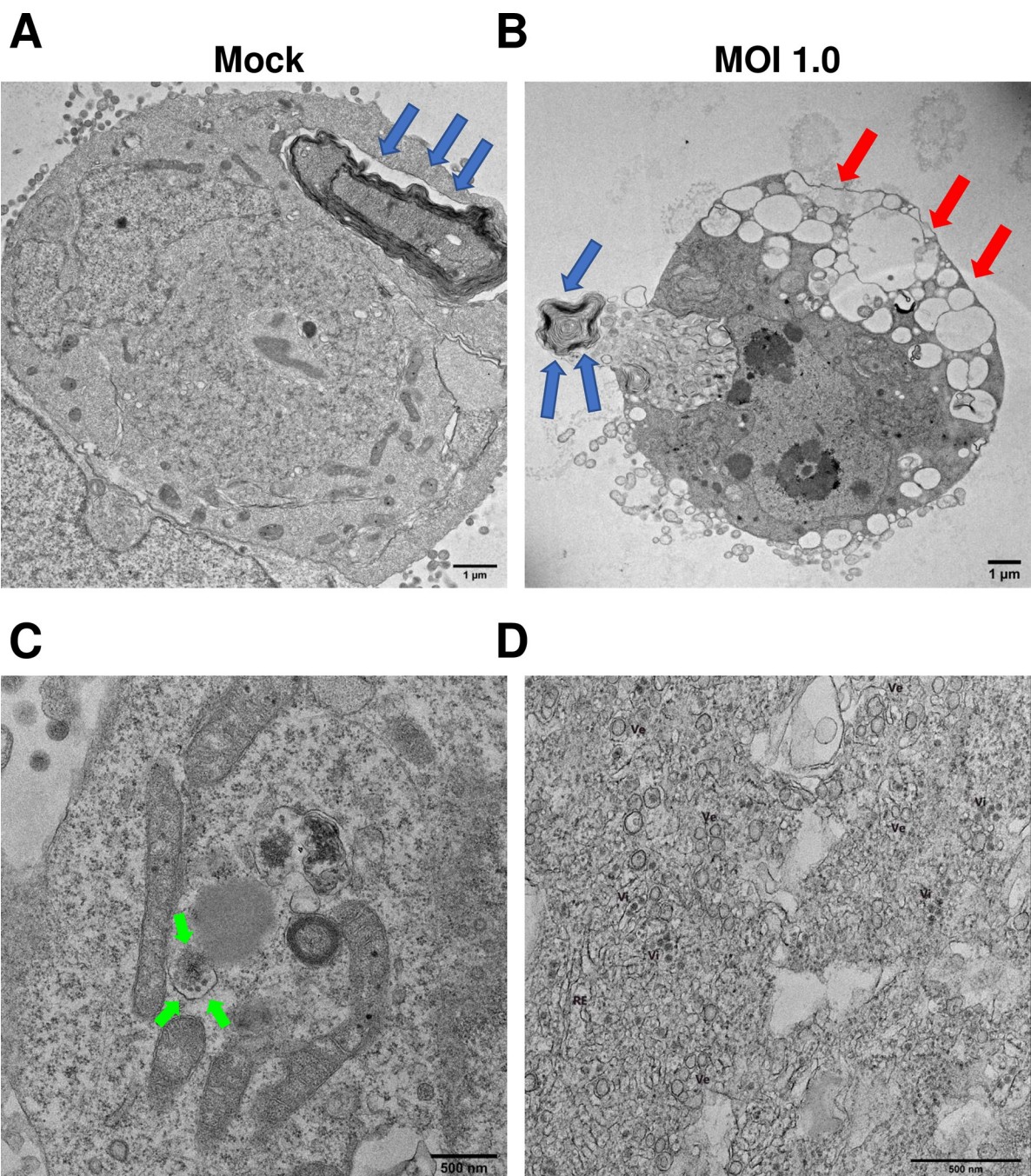

**Fig 6. Virions generated after ZIKV infection are concentrated in vesicles.** The astrocytes-hNS1 culture was infected with ZIKV (MOI of 1) or mock-infected. At 6 dpi, the monolayers were fixed and processed for observation by transmission electron microscopy. (A) Representative micrograph of an entire astrocyte-Hns1. Cells are characterized by large multilamellar bodies (see blue arrows). (B) ZIKV-infected cells at 6dpi showed many vacuoles (see red arrows) and the multilamellar bodies were very rare in infected cellsand were small (see blue arrows). (C) ZIKV-infected cells, where virions delimited in vesicles are observed (see green arrows). (D) ZIKV-infected cells show multiple virions associated with vesicular bodies and close to the endoplasmic reticulum.

(Fig 7D). Upon quantification of the number of mitochondria per cell, we did not find significant differences upon infection, 21 and 15 (mock infected day 3 and 6, respectively), 27 (3 dpi) and 21 (6 dpi) (Fig 7E). However, the area of the mitochondria was significantly different,

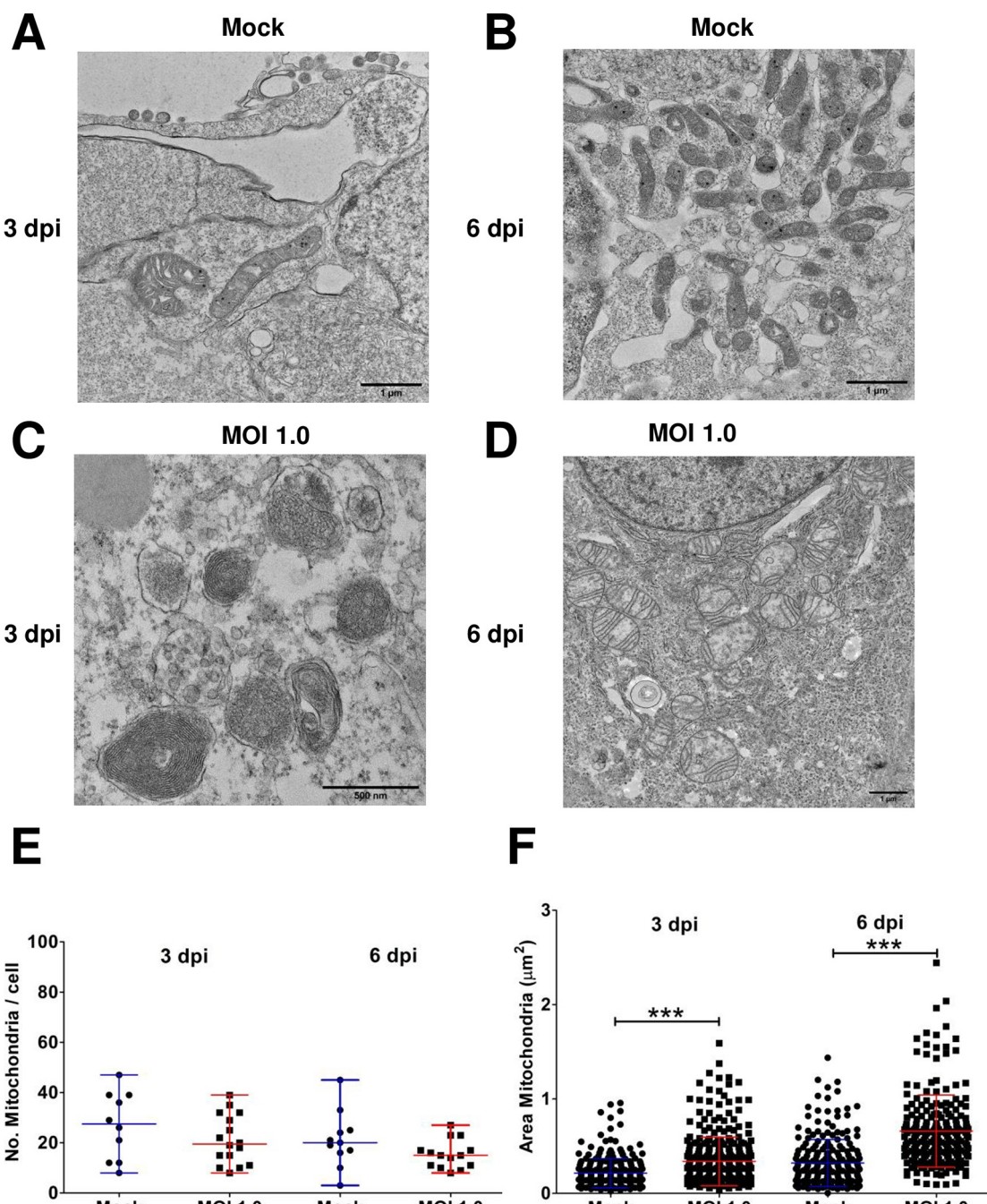

**Fig 7. ZIKV infection in astrocytes-hNS1 causes changes in the mitochondria.** The astrocyte-hNS1 culture was infected with ZIKV (MOI of 1) or mock-infected. The monolayers were fixed at 3 and 6 dpi and processed for observation by transmission electron microscopy. (A) Mock-infected cells at three dpi. (B) Mock-infected cells at six dpi. The mitochondria of the mock-infected cells show the homogeneous size and parallel cristae. (C) ZIKV-infected cells during 3 dpi showing mitochondria with concentric cristae and mitochondria with the sparse mitochondrial matrix. (D) ZIKV-infected cells 6 dpi showed decreased mitochondrial cristae and lighter electron density. (E) Quantitative analysis of the number of mitochondria per cell, the mean and standard deviation are presented, p = 0.06, ANOVA. (F) Quantitative analysis of the size of the mitochondria expressed in the mitochondrial area ($\mu m^2$), the median and range are presented, p<0.0001, Kruskal Wallis test with a post hoc Dunn's Multiple Comparison Test. In order to compare the area of the mitochondria we only analyzed organelles with the same orientation.

progressively increasing with infection: uninfected cells (0.21 ± 0.15 μm$^2$), 3 dpi
(0.34 ± 0.26 μm$^2$, p<0.0001) and 6 dpi (0.66 ± 0.38 μm$^2$, p<0.0001) (Fig 7F). These results
strongly suggest mitochondria damage and loss of function upon ZIKV infection.

## Lipid droplets are absent in infected astrocytes

Lipid-droplets (LD) are membraneless organelles that derive from the endoplasmic reticulum
and serve to storage lipids [53]. There is evidence that LDs contribute to the assembly of the
progeny of some flaviviruses, including DENV and hepatitis C virus [54]. We analyzed LDs in
ZIKV-infected astrocytes-hNS1 at 3 and 6 dpi. There were no differences in the morphology
of the LD between the mock-infected and infected cells at 3 dpi; however, the electron density
of the LD in infected cells seemed lower than in the mock-infected cells (Fig 8A to 8C). Strik-
ingly, by 6 dpi, we could not find LDs in any of the infected cells (Fig 8D). When we quantified
the number of LDs, we noticed that there was an increase in the number of LDs at 3 dpi: unin-
fected cells (3.4 ± 1.2 LDs/cell) and ZIKV-infected cells (28.5 ± 11.4 LDs/cell, p<0.001) (Fig
8E). By day 6 pi LDs were absent in infected cells: uninfected cells (6.9 ± 4.3 LDs/cell) and
infected cells (0 LDs/cell, p = 0.0058) (Fig 8F). We did not observe changes in the area of the
LDs in the infected cells, nor did we observe viral replication associated with these organelles.
Altogether we show evidence that astrocytes made from neural progenitor cells are a conve-
nient model for studying the damage that ZIKV infection causes in the central nervous system.

## Discussion

ZIKV can infect the developing brain when it crosses the placental and blood-brain barriers.
Therefore, ZIKV has been associated with problems in the development of the CNS such as
microcephaly, cerebral calcifications, ventriculomegaly, and thin cerebral cortex [8, 11, 12, 53].
In the developing brain, ZIKV can infect NPCs, neurons, and astrocytes. Recent studies have
shown that ZIKV infection can persist for at least a month in primary cultures of human astro-
cytes [28]. Taking this into account, it is plausible that persistently infected astrocytes play an
essential role in generating neuronal damage.

In this study, we generated a culture of astrocytes from NPCs. Once we characterized this
cell line and determined that it has a correct karyotype and a gene expression profile consistent
with glial cells, we decided to analyze the permissibility to ZIKV infection. We show that astro-
cytes-hNS1s are permissible to ZIKV, and that infection compromises cell survival, measured
indirectly through decreased metabolic activity, increased ROS production and gross cyto-
pathic changes, confirming results from another group [27]. We observed about 26% of
infected cells at 3 dpi (MOI of 1), which is also similar to a previous study using cultures of
human fetal cortical astrocytes infected with clinical isolates of ZIKV [55].

Astrocytes are critical for CNS homeostasis, particularly for the regulation of glutamatergic
neurotransmission. We believe that the observation that ZIKV infection decreases *EAAT1* and
*GS* expression is relevant, and may indicate possible changes in glutamate regulation in the
intracellular space. Other viruses also induce neuronal damage by compromising the homeo-
stasis of the glutamatergic system. For instance, human immunodeficiency virus type 1 (HIV-
1) infection of fetal cortical astrocytes can decrease the expression of EAAT1 and EAAT2 at
the messenger and protein levels, and this alters the uptake of $^3$H-D-aspartate [56]. The
reduced expression of *EAAT1* and *GS* has not been previously reported for ZIKV. Our data
call to further investigate whether glutamate uptake is modified and whether this could gener-
ate neurons death due to an excess of glutamate.

In a study with isolated rodent cortical neurons, ZIKV infections altered the functionality
and expression of NMDA$_R$ subunits [57]. Olmo et al. found overexpression of NMDA$_R$

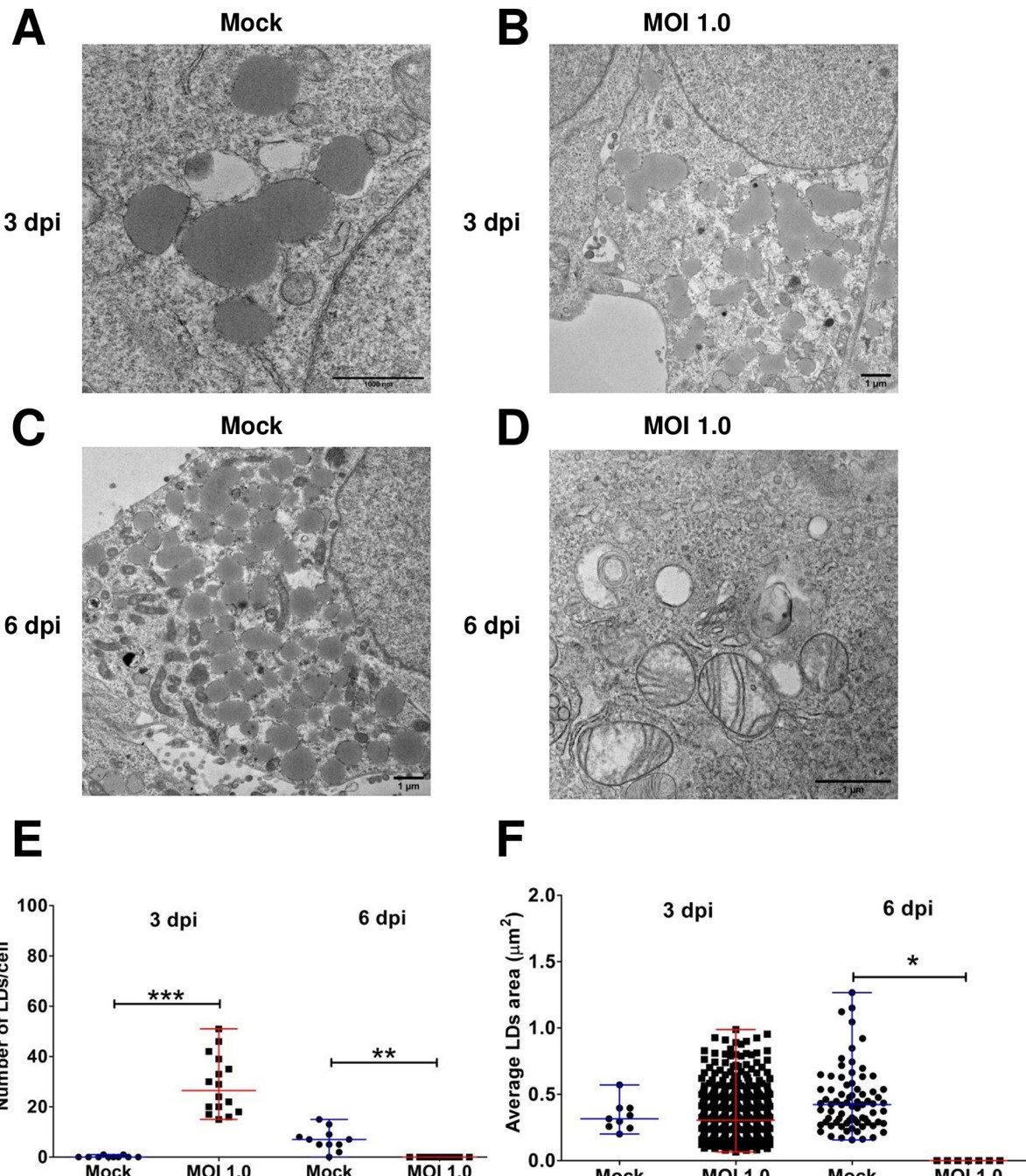

**Fig 8. ZIKV infection in astrocytes-hNS1 causes changes in lipid droplets (LDs).** The astrocyte-hNS1 culture was infected with ZIKV (MOI of 1) or mock-infected. At 3 and 6 dpi, the monolayers were fixed and processed for observation by transmission electron microscopy. (A) Mock-infected cells at three dpi. (B) Mock-infected cells at six dpi respectively, LDs can be observed in the cell's cytoplasm. (C) ZIKV-infected cells (MOI of 1) at three dpi showed many LDs throughout the cytoplasm. (D) Representative micrograph of a ZIKV-infected cell at six dpi showing abnormal mitochondria and a complete absence of LDs. (E) Quantitative analysis of the number of LDs /cells, median, and range are presented, **$p<0.001$, ***$p<0.0001$, U Mann Whitney test. (F) Quantitative analysis of LDs size (area, $\mu m^2$), the mean and standard deviation are presented, $p = 0.10$, Kruskal Wallis test, comparisons between the groups where measurements of LDs area could be determined.

subunit 2 (GLUN2) at 24 hours, accompanied by an increase in the concentration of intracellular calcium and extracellular glutamate. We also found that ZIKV infection increases $NMDA_R$ expression. Our results open the door to investigate whether there are changes in intracellular calcium due to the activation of $NMDA_R$ and whether it is accompanied by cell death, as described before [58, 59]. Future studies should also address the expression of other $NMDA_R$ subunits at the mRNA and protein levels.

We also observed a reduced *GFAP* expression. This data differs from the literature where ZIKV produces glial activation, demonstrated by increased *GFAP* expression [27–29]. However, another study reports a decrease in GFAP expression at the protein level when cells are infected with ZIKV [60]. Huang et al. (2018) analyzed ZIKV infection in astrocytes and found that many infected cells do not express *GFAP*. They found that infected cells that do not express *GFAP* express *S100-B* (the calcium-binding protein B). The overexpression of *S100-B* demonstrated glial activation [35, 61, 62]. It would be interesting to analyze whether the expression of S100-B is modified in ZIKV-infected astrocytes-hNS1.

ZIKV, like other flaviviruses, contains phosphatidylserine in the viral envelope, which is taken from the infected cell during viral egress [63]. The incorporation of phosphatidylserine on the viral envelope facilitates viral adsorption and internalization by TAM-mediated phagocytosis. The TAM family of receptors (*e.g.*, Tyro-3, Axl, and Mertk) recognizes the phosphatidylserine present in apoptotic cell membranes, promoting cell phagocytosis. We found that astrocytes-hNS1 expressed all three genes, perhaps explaining their permissibility to ZIKV infection. The gene expression of TAM receptors is modified at five dpi. On the one hand, AXL and Mertk are overexpressed. Ojha and collaborators analyzed the expression of TAM receptors in primary cultures of astrocytes. They found that AXL was overexpressed, and that blockade of this receptor decreased viral infection [29]. The expression of AXL in astrocytes-hNS1 is consistent with reports showing that infection in astrocytes is mediated by AXL [26, 29]. On the other hands, we found that ZIKV infection decreases tyro-3 expression. The expression of tyro-3 is consistent with a study in placentas, where Tyro-3 expression was decreased [64]. However, other studies have shown that Tyro-3 is increased in astrocytes [29]. These discrepancies could be due to multiple factors, among the most important, the viral strains, the number of passages to obtain the virus, the origin of astrocytes, the purity of the culture, the MOI used and the time of infection.

The PPARγ gene is a regulator of lipid metabolism that participates in adipogenesis [65]. Different cellular factors necessary for flavivirus replication are involved in the metabolism of lipids [54, 66]. We did not see changes in the expression of *PPARγ* after infection. Overexpression of PPARγ has been previously reported in NPCs infected with ZIKV [32, 65].

The apolipoprotein E (APOE) is a highly expressed protein with a central role in lipid metabolism in neural cells [67, 68], so we were interested in knowing if there was a relationship between APOE expression and ZIKV production in astrocytes. We report that there are no changes at the transcriptional level of this gene. Our thin-section TEM data clearly shows that upon infection there is an alteration of the lipid metabolism; we observed a decrease in the number of large multilamellar bodies, and an increase followed by a decrease in the number of LD. Therefore, in our culture of astrocytes, other genes must be participating in regulating the metabolism distribution of lipids that help the production of ZIKV viral particles.

Finally, we demonstrate, by thin-section TEM, how ZIKV infection structurally modifies hNS1-derived astrocytes. The first approach we carried out was to analyze the changes in the mitochondria, finding that the number of mitochondria per cell was maintained in ZIKV-infected and uninfected cells. However, when analyzing the size of the mitochondria (mitochondrial area, $\mu m^2$), we found that the mitochondrial area greater in infected cells at both three and six dpi than in the controls. We also found that the integrity of the cristae and their

organization is affected by the infection. This could suggest a possible mitophagy process, which has been previously observed in ZIKV [69] and classical swine fever virus (CSFV) [70]. Mitochondria play a fundamental role in the energy supply for the cell. In the infected condition, the mitochondria will be overloaded, as the energy requirements for cell survival and virion production imply energy use. An increase in ROS can be presented by mitochondrial overload. Damage to mitochondria has been analyzed in various studies. For example, by using astrocytes derived from NPCs and infecting them with ZIKV, it was shown that the infection could increase ATP synthesis at 18 hpi, but the ATP reserve capacity is reduced to 24, 36, and 48 hpi. Furthermore, at 48 hpi, a decrease in mitochondrial respiration routine is observed due to mitochondrial damage produced by ROS [27]. In another study, using ZIKV-infected human retinal epithelial cells, a change in mitochondrial morphology was observed by confocal microscopy and changes in mitochondrial membrane potential, suggesting that ZIKV infection induces an imbalance of fusion/fission of mitochondria [71].

Interestingly, at three dpi the number of LDs per cell was higher in ZIKV-infected than in the mock-infected cells. However, when looking at cells at six dpi, LDs were absent in infected cells. The area of the LDs was similar in both infected and uninfected cells at three dpi and in uninfected cells at six dpi. The dynamics with which these cellular compartments appear to increase and ultimately decrease tell us about lipids influence on the ZIKV viral cycle. García et al. (2020) used Huh-7 cells to characterize changes in LDs at 24 hpi and found that the number of LDs/cell was higher in uninfected cells compared to ZIKV-infected cells. Furthermore, a decreased volume of LDs was observed in ZIKV-infected cells compared to uninfected cells [71]. We believe that the decrease in LDs in astrocytes occurs due to the production of viral particles since the replication of some flaviviruses, for example, DENV, depends on the β-oxidation of triglycerides present in LDs [72, 73].The use of lipid droplets as lipid substrates could explain why in our work, the amount of LD decreases at higher times of infection (6 dpi), which coincides in being the post-infection time in which more viral particles and a lower amount of LD.

Research on cellular factors necessary for viral replication, such as lipids, offers information for the development of treatments that could prevent the generation of microcephaly and other malformations associated with ZIKV infection in pregnancy. Given the great importance of lipids in the viral production cycle, it could look at the potential of agents that inhibit lipid biosynthesis and have been shown not to interfere with pregnancy, for example, metformin [54].

In summary, we developed a culture of astrocytes that were derived and purified from pluripotent cells. The astrocytes-hNS1 were characterized by analyzing the expression of the GFAP marker and the gene expression of GFAP, EAAT1, EAAT2, and GS. This culture maintained a stable karyotype. Furthermore, astrocytes-hNS1 are susceptible and permissive to ZIKV infection. This viral infection leads to ROS production and a decrease in cell viability; the gene expression for GFAP, EAAT1, and GS is downregulated while for NMDA$_R$ increased. This suggests that there are changes that compromise the homeostasis of neuronal communication, specifically communication by glutamatergic neurotransmission. The overexpression of Mertk and AXL, and the downregulation of Tyro-3 suggest that these are receptors that promote ZIKV infection in our culture. Finally, we found changes in the size of the mitochondria and a decrease in the number of LDs in the cells, which helps us better understand virion production and propose experiments that help develop therapies to mitigate the effects of ZIKV infection.

The data presented here show the overall effect of ZIKV in infected cells; however, in order to determine if changes in cell morphology and biochemistry are the result of an early inflammatory process or a consequence of the cytopathic effects, we would like to perform blocking

experiments that help us to separate the effects of these processes. However, blocking ZIKV entry is a complex experimental problem. There are multiple candidate receptors involved in ZIKV infection [74]. In cells of the nervous system the expression of at least four proteins has been associated with ZIKV entry: DC-SING [75], Tyro-3 [76], AXL [26, 77], and Mertk [76]. Although carrying out the blocking experiments of these receptors would represent a greater understanding of the mechanism underlying the viral infection, these methodological approaches are beyond the scope of our work, so it could be a perspective for addressing in the future. As it has been shown by others [26, 64, 74, 77] blocking one receptor at a time is not enough to complexly block entry; hence, in order to completely inhibit viral entry, we would have to block the interaction of the virus with multiple receptors. Furthermore, given the role of these proteins in the biology of neural cells it is possible that blocking these receptors could result in profound cellular alterations that could hamper our understanding of the effects of the viral infection on these cells.

Although the question remains whether ZIKV will ever be able to cross the adult blood-brain barrier and affect the brain, astrocytes are highly susceptible to ZIKV, posing a high risk, especially for pregnant women. Astrocytes control redox homeostasis, help control neuronal communication, and maintain the blood-brain barrier in optimal conditions. A decrease in the number of astrocytes can lead to motor and cognitive problems and neuronal loss without the need for ZIKV to infect neuronal cells. Therefore, it would be worth studying the infection of this type of cells in depth to generate drug therapies in the future.

## Supporting information

**S1 Fig. Gene expression of glial cell marker GFAP in cultured astrocytes-hNS1 by One-step RT-PCR.** The astrocytes-hNS1 culture was seeded. At three days of proliferation, cellular RNA was recovered by analyzing the expression of gene markers of glial cells. Expression of GFAP in cells astrocytes-hNS1 and hNS1 cell line.
(TIF)

**S1 Table. The primers used for PCR.** Primers used and their characteristics.
(DOCX)

## Acknowledgments

EI.R.H (CVU: 775486) and M.C.S. (CVU: 714882) received scholarships from CONACYT.

M. Sc Arleth Miranda Lopez for her advice on the karyotype technique.

Dr. Fernando Peña Ortega, Dra. Adriana Monsiváis Urenda, and Dr. Christian García Sepulveda by suggestions during the PhD of EI.R.H.

Dr. Alberto Martínez-Serrano ("Severo Ochoa" Molecular Biology Center, Madrid, Spain) for donating the hNS-1 cell line.

Dr. Ezequiel Fuentes Pananá by suggestions and revision of original paper.

## Author Contributions

**Conceptualization:** Edson Iván Rubio-Hernández, Mauricio Comas-García, Claudia G. Castillo.

**Formal analysis:** Edson Iván Rubio-Hernández.

**Funding acquisition:** Mauricio Comas-García, Claudia G. Castillo.

**Investigation:** Edson Iván Rubio-Hernández, Miguel Angel Coronado-Ipiña, Mayra Colunga-Saucedo.

**Methodology:** Edson Iván Rubio-Hernández, Mauricio Comas-García, Miguel Angel Coronado-Ipiña, Mayra Colunga-Saucedo, Hilda Minerva González Sánchez, Claudia G. Castillo.

**Project administration:** Mauricio Comas-García, Claudia G. Castillo.

**Resources:** Mauricio Comas-García, Claudia G. Castillo.

**Supervision:** Mauricio Comas-García, Claudia G. Castillo.

**Validation:** Edson Iván Rubio-Hernández.

**Visualization:** Edson Iván Rubio-Hernández.

**Writing – original draft:** Edson Iván Rubio-Hernández.

**Writing – review & editing:** Mauricio Comas-García, Hilda Minerva González Sánchez, Claudia G. Castillo.

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
