## [Decision Letter · Decision Letter 0]

7 Jan 2023

PONE-D-22-28463Astrocytes derived from Neural Progenitor Cells are susceptible to Zika Virus Infection.PLOS ONE

Dear Dr. Castillo,

Thank you for submitting your manuscript to PLOS ONE. After careful consideration, we feel that it has merit but does not fully meet PLOS ONE’s publication criteria as it currently stands. Therefore, we invite you to submit a revised version of the manuscript that addresses the points raised during the review process.

We look forward to receiving your revised manuscript.

Kind regards,

Abhishek Kumar Singh, Ph.D.

Academic Editor

PLOS ONE

Journal Requirements:

Reviewers' comments:

Reviewer's Responses to Questions

**Comments to the Author**

1. Is the manuscript technically sound, and do the data support the conclusions?

Reviewer #1: Yes

Reviewer #2: Yes

2. Has the statistical analysis been performed appropriately and rigorously? 

Reviewer #1: No

Reviewer #2: Yes

3. Have the authors made all data underlying the findings in their manuscript fully available?

Reviewer #1: Yes

Reviewer #2: Yes

4. Is the manuscript presented in an intelligible fashion and written in standard English?

Reviewer #1: Yes

Reviewer #2: Yes

5. Review Comments to the Author

Reviewer #1: The manuscript by Rubio-Hernadez et al used astrocytes derived from the human neural progenitor cell line (hNS1) to study Zika virus infection. They found that these astrocytes expressed several astrocyte marker genes and maintained a stable karyotype. More importantly, consistent with the existing literature, these astrocytes are susceptible to Zika virus infection. This viral infection leads to ROS production and a decrease in cell viability, as well as decrease in gene expression of GFAP, EAAT1, and GS. The viral infection also leads to increase of two Zika virus entry receptors (Mertk and AXL), along with mitochondria and lipid droplet changes. Overall, although astrocytes have previously been found to be susceptible to Zika virus infection, the susceptibility of astrocytes derived from the hNS1 cell line to Zika virus infection appears to be new and could become a valid cellular model for Zika virus infection. Subsequent characterizations of various morphological, cellular, and subcellular changes of astrocytes after Zika virus are extensive but mostly expected phenotypes. Furthermore, without blocking experiments, these changes in astrocyte gene expression, mitochondrial morphology, viral entry receptors, and lipid droplets are all association but not necessarily providing any mechanistic insight. These are the major concerns that need to be addressed.

Major points:

1. The authors referred the astrocyte cultures as primary cultures in the manuscript. Primary culture typically refers to the culturing of cells directly obtained from a multicellular organism. Since the astrocytes in the current study were derived from the hNS1 cell line, it is not appropriate to call them primary cultures.

2. In Fig. 3. Statistical analysis appears to be performed on normalized control group, which is incorrect. Please clarify.

3. Changes in astrocyte gene expression, ROS, mitochondrial morphology, viral entry receptors, and lipid droplets are likely all pointing to early inflammatory and later cytopathic effects. Without any blocking experiments, it is difficult to conclude whether these identified changes will provide any therapeutic benefits.

Minor points:

1. More editing is required to improve the quality of this manuscript. e.g., Page 14, Line 319: grammatical error “Twenty first days after the differentiation”; Figs 1-2, all scale bars need higher resolutions.

2. In Pages 14-15, the authors cited reference #4 for astrocyte and neuronal differentiation of the NS1 cell line. However, the reference #4 is a review paper with no information on such data.

3. Page 14, Line 315, although astrocytes are now known to be required for the proper generation of the myelin sheath, astrocytes do not form myelin.

Reviewer #2: The manuscript of Castillo et al. is an interesting experimental in vitro study focused on the primary human astrocytes from hNS1 cells (astrocytes hNS1), characterized by the expression of bona fide astrocyte markers. These findings significantly increase our understanding of the pathogenic mechanisms of ZIKV infection in the CNS and provide the foundation of a model for further characterization. It is a planned study and the data generated look convincing, and the findings of the study may be highly important in the research field. ZIKV infection in this cell type would be worth studying in-depth to generate pharmacological therapies.

6. PLOS authors have the option to publish the peer review history of their article (what does this mean?). If published, this will include your full peer review and any attached files.

Reviewer #1: No

Reviewer #2: No

---

## [Author Response · Author response to Decision Letter 0]

2 Feb 2023

Ref.: PONE-D-22-28463

Astrocytes derived from Neural Progenitor Cells are susceptible to Zika Virus Infection.

Plos One

Response to Reviewer’s Concerns:

We thank the reviewers for their observations. We have addressed each concern in this response and the revised manuscript. The comments by the reviewer are in bold type, followed by the answers in regular type. The changes within the manuscript were uploaded as a separate file labeled ‘Revised Manuscript with Track Changes’.

Reviewer #1: The manuscript by Rubio-Hernandez et al. used astrocytes derived from the human neural progenitor cell line (hNS1) to study Zika virus infection. They found that these astrocytes expressed several astrocyte marker genes and maintained a stable karyotype. More importantly, consistent with the existing literature, these astrocytes are susceptible to Zika virus infection. This viral infection leads to ROS production and a decrease in cell viability, as well as decrease in gene expression of GFAP, EAAT1, and GS. The viral infection also leads to increase of two Zika virus entry receptors (Mertk and AXL), along with mitochondria and lipid droplet changes. Overall, although astrocytes have previously been found to be susceptible to Zika virus infection, the susceptibility of astrocytes derived from the hNS1 cell line to Zika virus infection appears to be new and could become a valid cellular model for Zika virus infection. Subsequent characterizations of various morphological, cellular, and subcellular changes of astrocytes after Zika virus are extensive but mostly expected phenotypes. Furthermore, without blocking experiments, these changes in astrocyte gene expression, mitochondrial morphology, viral entry receptors, and lipid droplets are all association but not necessarily providing any mechanistic insight. These are the major concerns that need to be addressed.

The present study focuses on obtaining an astrocyte culture from human neural progenitor cells that are characterized by bona fide expression of astrocyte markers. Once the astrocyte culture was obtained and characterized, we studied the effect of ZIKV infection in this. This study used a mock infection control (supernatant of uninfected Vero E6 cells). This control allows us to determine that the changes we report (ROS, changes in mitochondria, and lipid drops) are a consequence of the viral infection. As the reviewer rightly mentions, these could be the direct or indirect results of ZIKV infection. However, for the moment, our objectives focused on analyzing whether our astrocytes were susceptible to ZIKV infection, to propose them as an in vitro system to investigate ZIKV infection in this cell type. 

Major points:

1. The authors referred the astrocyte cultures as primary cultures in the manuscript. Primary culture typically refers to the culturing of cells directly obtained from a multicellular organism. Since the astrocytes in the current study were derived from the hNS1 cell line, it is not appropriate to call them primary cultures.

These changes were made. We modified the manuscript in lines 86, 95, and 325.

2. In Fig. 3. Statistical analysis appears to be performed on normalized control group, which is incorrect. Please clarify.

In figure no. 3, the determination of the metabolic activity and the oxygen species production is expressed as arbitrary fluorescence units, and the absolute fluoresce value can change between experiments. This is the reason why in every experiment, we had a mock-infected group so that arbitrary fluorescence units were set up to 100%. In other words, each of the independent experiments was normalized with its respective mock-infected control. Because the data are expressed in percentages, the data were analyzed to determine whether there are significant differences between the groups (Mock, MOI 0.1, MOI 1.0, and H2O2 control) using the Kruskal Wallis test with Dunn's post hoc test. To avoid confusion, remove the line indicating 100%, and we add the deviation data from the mock infection group.

3. Changes in astrocyte gene expression, ROS, mitochondrial morphology, viral entry receptors, and lipid droplets are likely all pointing to early inflammatory and later cytopathic effects. Without any blocking experiments, it is difficult to conclude whether these identified changes will provide any therapeutic benefits.

All the experiments were performed with is respective control group (mock-infected group), and all comparisons are made with respect to the mock-infected control. Therefore, because of these comparisons, it is clear that the reported changes are a consequence of the viral infection. The changes in ZIKV-infected cells we have characterized are a product of a viral infection, and thus they also represent those from the early inflammatory and later cytopathic effects.

It is important to point out that multiple candidate receptors are involved in ZIKV entry (Agrelli, de Moura, Crovella, & Brandão, 2019). In cells of the nervous system, this entry process has been associated with at least the expression of four transmembrane proteins: DC-SING (Deiva et al., 2006), Tyro-3 (Ji et al., 2014), AXL (Chen et al., 2018; Jimenez et al., 2021), and Mertk (Ji et al., 2014). The blocking experiments suggested by the reviewer represent by themselves a series of complex experimental designs that merit several articles. Furthermore, it is very important to point out that it has been shown that because there are multiple receptors for viral entry, blocking and silencing a particular receptor is not enough to completely inhibit the viral infection.

Finally, our goal was to analyze the effects of Zika virus infection on metabolic activity, ROS, and ultrastructural changes of mitochondria and lipid droplets, without considering, for the time being, the objective of developing therapeutic targets. Therefore, although the reviewer’s suggestion is very interesting, it is outside the scope of this article, and they will also merit their own articles. We added a note in the manuscript at lines 696-713 and 723-730.

Minor points:

1. More editing is required to improve the quality of this manuscript. e.g., Page 14, Line 319: grammatical error “Twenty first days after the differentiation”; Figs 1-2, all scale bars need higher resolutions.

We Revised and corrected the manuscript. Twenty first days were changed to twenty-one days at line 326.

In figures 1 and 2, the correspondence in microns of all bars was removed. The equivalence of the scale bar in micrometers was placed at the figure legends (Figure 1, line 350; Figure 2, line 380 and line 390).

We revised and the adequate changes at this manuscript were made.

2. In Pages 14-15, the authors cited reference #4 for astrocyte and neuronal differentiation of the NS1 cell line. However, the reference #4 is a review paper with no information on such data.

We reviewed and corrected the manuscript. Indeed, there was an error in the reference. The correct references are 39 and 40 at line 330.

3. Page 14, Line 315, although astrocytes are now known to be required for the proper generation of the myelin sheath, astrocytes do not form myelin.

We made changes to the manuscript. We changed ‘form myelin’ to ‘Promote myelination’ at line 321.

Reviewer #2: The manuscript of Castillo et al. is an interesting experimental in vitro study focused on the primary human astrocytes from hNS1 cells (astrocytes hNS1), characterized by the expression of bona fide astrocyte markers. These findings significantly increase our understanding of the pathogenic mechanisms of ZIKV infection in the CNS and provide the foundation of a model for further characterization. It is a planned study and the data generated look convincing, and the findings of the study may be highly important in the research field. ZIKV infection in this cell type would be worth studying in-depth to generate pharmacological therapies.

We appreciate the comments, and the culture of astrocytes derived from human neural progenitor cells could be a handy tool to investigate ZIKV neuroinfections further.

REFERENCES:

Agrelli, A., de Moura, R. R., Crovella, S., & Brandão, L. A. C. (2019). ZIKA virus entry mechanisms in human cells. Infection, Genetics and Evolution, 69, 22-29. doi:https://doi.org/10.1016/j.meegid.2019.01.018

Chen, J., Yang, Y. F., Yang, Y., Zou, P., Chen, J., He, Y., . . . Xu, J. (2018). AXL promotes Zika virus infection in astrocytes by antagonizing type I interferon signalling. Nat Microbiol, 3(3), 302-309. doi:10.1038/s41564-017-0092-4

Deiva, K., Khiati, A., Hery, C., Salim, H., Leclerc, P., Horellou, P., & Tardieu, M. (2006). CCR5-, DC-SIGN-dependent endocytosis and delayed reverse transcription after human immunodeficiency virus type 1 infection in human astrocytes. AIDS Res Hum Retroviruses, 22(11), 1152-1161. doi:10.1089/aid.2006.22.1152

Ji, R., Meng, L., Jiang, X., Cvm, N. K., Ding, J., Li, Q., & Lu, Q. (2014). TAM Receptors Support Neural Stem Cell Survival, Proliferation and Neuronal Differentiation. PLoS One, 9(12), e115140. doi:10.1371/journal.pone.0115140

Jimenez, O. A., Narasipura, S. D., Barbian, H. J., Albalawi, Y. A., Seaton, M. S., Robinson, K. F., & Al-Harthi, L. (2021). β-Catenin Restricts Zika Virus Internalization by Downregulating Axl. J Virol, 95(17), e0070521. doi:10.1128/jvi.00705-21

---

## [Editor Report · Decision Letter 1]

8 Mar 2023

Astrocytes derived from Neural Progenitor Cells are susceptible to Zika Virus Infection.

PONE-D-22-28463R1

Dear Dr. Castillo,

We’re pleased to inform you that your manuscript has been judged scientifically suitable for publication and will be formally accepted for publication once it meets all outstanding technical requirements.

Kind regards,

Abhishek Kumar Singh, Ph.D.

Academic Editor

PLOS ONE
---

## [Editor Report · Acceptance letter]

20 Mar 2023

PONE-D-22-28463R1 

Astrocytes derived from Neural Progenitor Cells are susceptible to Zika Virus Infection. 

Dear Dr. Castillo:

I'm pleased to inform you that your manuscript has been deemed suitable for publication in PLOS ONE. Congratulations! Your manuscript is now with our production department. 

Kind regards, 

on behalf of

Dr. Abhishek Kumar Singh 

Academic Editor

PLOS ONE